# Cohort profile: the Mendelian randomisation in pregnancy (MR-PREG) collaboration – improving evidence for prevention and treatment of adverse pregnancy and perinatal outcomes

Nancy McBride,[1,2] Gemma L Clayton [1,2] Ana Goncalves Soares [1,2] Qian Yang,[1,2,3,4] Tom A Bond [1,2,5] Amy Taylor,[1,2,6] Charikleia Chatzigeorgiou,[1,2] Elisabeth Aiton [1,2] Jane West [7,8] Maria C Magnus,[9] Deborah A Lawlor,[1,2] Maria Carolina Borges [1,2]

NM, GC and AGS are joint first authors.

For numbered affiliations see end of article.

**Correspondence to**
Dr Gemma L Clayton;
gemma.clayton@bristol.ac.uk

## ABSTRACT

**Purpose** Adverse pregnancy and perinatal outcomes (APPOs), including pre-term birth, pre-eclampsia and gestational diabetes, can result in maternal and neonatal morbidity and mortality, parental anxiety and increased healthcare costs. A better understanding of the causes of APPOs is essential to inform lifestyle and pharmaceutical interventions for their prevention and management. Given the difficulty of undertaking randomised controlled trials in pregnant women, triangulating evidence from across methods with different sources of bias may improve causal inference for APPOs. The purpose of the Mendelian randomisation in pregnancy (MR-PREG) collaboration is to support such triangulation using genetic (eg, Mendelian randomisation (MR)) and non-genetic (eg, partner negative controls) approaches to investigate the causal effects of maternal exposures on a comprehensive set of APPOs.

**Participants** The MR-PREG collaboration includes individual participant data from three birth cohorts (two from the UK and one from Norway) and UK Biobank, as well as summary data from FinnGen and publicly available genome-wide association studies (GWAS). Data have been harmonised across studies and currently include information on up to 35 APPOs in up to 707 797 women.

**Findings to date** The main aims of MR-PREG are to strengthen the evidence base for (1) prevention, by advancing understanding of maternal lifestyle factors on APPOs, (2) the role of pre-conceptional health, by improving understanding of the effect of maternal pre-existing conditions on APPOs, and (3) treatments, by evaluating the efficacy and safety of existing medications used for pre-existing conditions, and by identifying and testing novel or repurposed therapies for APPOs. To date, our published work has mainly addressed aims 1 and 3. Examples include triangulation of evidence from MR, conventional multivariable regression and paternal negative control, showing that

## STRENGTHS AND LIMITATIONS OF THIS STUDY

⇒ We have curated data on 35 adverse pregnancy and perinatal outcomes (APPOs) and harmonised data across multiple studies to support large-scale investigations of causes of APPOs.

⇒ The breadth and nature of the data support triangulation of evidence from a range of genetic and non-genetic methods, each with distinct sources of bias, to identify causes of APPOs.

⇒ Over the coming year, we will further enhance the data to enable identification of molecular mechanisms underlying APPOs.

⇒ The current curated dataset has limited power to detect causal effects for rarer APPOs, such as congenital anomalies and low Apgar scores at 5 min, particularly when using genetic methods. Future work will incorporate larger samples and rare genetic variant data.

⇒ Participants are predominantly of European ancestry; future efforts will focus on increasing ancestral diversity within the data.

higher maternal body mass index increases the risk of multiple APPOs, as well as the identification of maternal circulating metabolites and proteins that may influence birth weight.

**Future plans** Future priorities include increasing diversity within the MR-PREG collaboration by expanding representation of participants from non-European ancestries. We are also integrating molecular data, including circulating protein levels and placental transcriptomics, to better characterise the molecular mechanisms underlying APPOs. Additionally, we are using whole-exome and whole-genome sequencing to identify novel causal genes and to inform the prioritisation of candidate therapeutic targets for APPOs.

## WHY WAS THE COLLABORATION ESTABLISHED?

A substantial proportion of pregnant women experience adverse pregnancy and perinatal outcomes (APPOs), including miscarriage, stillbirth, gestational diabetes mellitus (GDM), gestational hypertension (GH), pre-eclampsia (PE), pre-term birth (PTB), or having a baby who is small (SGA) or large for gestational age (LGA). Globally, one in six recognised pregnancies results in miscarriage,[1] one in six women develops GDM,[2] one in 10 women experiences hypertension during pregnancy[3] and one in 10 babies is born preterm.[4] Some APPOs are major causes of severe morbidity and mortality for mothers and their babies.[5] For illustration, hypertensive disorders of pregnancy (HDP), including GH and PE, account for approximately 14% of all maternal deaths worldwide.[6] In addition, APPOs are associated with long-term adverse physical and mental health outcomes, as well as substantial costs to families, healthcare systems and society.[7 8] In the UK, the short-term costs of miscarriage alone are estimated to be around £471 million per year, reflecting costs to health services, families and loss of productivity.[1]

The marked variation in the prevalence of several APPOs across regions and over time, although influenced by different screening and diagnostic practices, indicates that many APPOs are at least partly preventable.[2] A better understanding of the underlying causes of APPOs is therefore crucial for guiding effective interventions to prevent them. Such knowledge would also enable antenatal care services to provide women and couples with accurate advice on the most plausible risk factors for reducing APPOs, potentially reducing the often-conflicting advice given to women about exposures that could impact their own and their babies' health during pregnancy. Furthermore, APPOs are often related, meaning that preventing one may help reduce the burden of others. For example, both GDM and HDP can influence fetal growth, leading to the delivery of LGA and SGA babies, respectively.[9 10] Preventing GDM and HDP is, therefore, a way of avoiding fetal over/under growth and the related complications.

In addition to lifestyle and health-related risk factors, there is also an urgent need to better understand the effects of pharmaceutical treatments during pregnancy. This includes therapies for the prevention and management of APPOs, as well as for the treatment of pre-existing conditions that are increasingly common among women of reproductive age, such as autoimmune, mental health, thyroid, reproductive and cardiometabolic disorders.[11–14] An increasing proportion of women start pregnancy on medication, and the use of medication among pregnant women has also been rising over the past few decades.[15] Despite this, pregnant women are rarely included in pre-licence randomised controlled trials (RCTs) largely due to concerns about potential teratogenic effects and uncertainty regarding drug dosing in the context of pregnancy-related physiological changes.[16 17] As a result, evidence on the benefits and risks of drugs for mother and baby is poor, investment in new drug development for APPOs is scarce and very few medications are explicitly licensed for use during pregnancy. This situation exposes mothers and babies to unknown risks of drugs, compels pregnant women and their doctors to undertreat medical conditions and forces them to make difficult decisions about whether to prescribe or continue medications and at what dose. It also means that APPOs, such as GDM and GH, are managed less well than equivalent conditions outside of pregnancy (ie, type 2 diabetes and hypertension).[16 17]

RCTs are the gold-standard method for testing the impact of lifestyle and pharmaceutical interventions to prevent or treat APPOs. For several years, national bodies have highlighted the need for clinical trials of medications in pregnant women, largely to no avail.[16 18] Reflecting this challenge, the UK 2021 Report of the Commission on Human Medicines Expert Working Group on Optimising Data On Medicines Used During Pregnancy recommended improved use, linkage and access to routinely collected healthcare data to support high-quality observational research and pharmacovigilance on the safety of medicines used during pregnancy and breastfeeding.[18]

In the absence of well-powered, well-conducted RCTs, we need to make the best use of observational data to improve the current evidence base on causes of APPOs and medication efficacy and safety during pregnancy. More than 20 years ago, the use of genetic variants to infer causal effects of exposures—a method known as Mendelian randomisation (MR)—was first proposed.[19] MR leverages the random allocation of genetic variants at conception to investigate the effects of modifiable risk factors on health outcomes. This method mitigates confounding by socioeconomic, behavioural or health-related factors that frequently affect traditional observational study analyses (figure 1).[19–21] Since then, its application has expanded significantly, including for assessing the effects of a few maternal risk factors on a small range of APPOs (eg,[22 2324]).

Building on this framework, MR has also been extended to evaluate the effects of medications through genetic proxies for drug targets (also referred to as drug target MR). The validity of drug target MR is supported by proof-of-concept studies comparing MR findings to RCT results for established medications, such as antihypertensives and statins.[25] Drug targets supported by genetic evidence have higher success rates and genetic evidence (including from MR) is increasingly used to prioritise (or de-prioritise) new drug targets to be tested in RCTs.[26–28]

As with all methods, MR is limited by violation of its assumptions. Triangulation of evidence acknowledges and exploits the fact that all methods have sources of bias.[29] It involves integrating multiple lines of evidence using different approaches (eg, different analytical methods, data sources or study designs) with distinct and unrelated sources of bias. If results are consistent, this increases the credibility of the estimated causal effect, as it is unlikely that different biases would produce similar results. Where there is disagreement across the different approaches, prior specification of key sources of bias and their direction for each approach can help determine

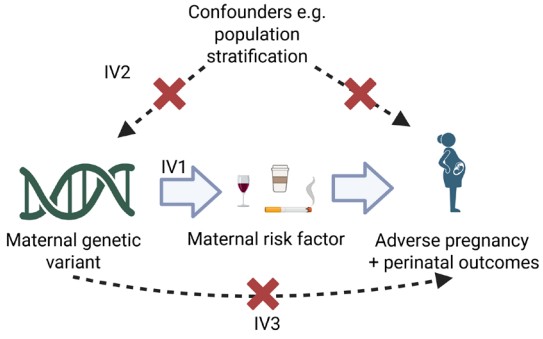

**Assumptions**

- The maternal genetic variant(s) are statistically robustly associated with the maternal risk factor during pregnancy **[relevance assumption, IV1]**
- There is no confounding of the maternal genetic variant(s) and adverse pregnancy and perinatal outcomes (APPOs) (i.e., population level confounders such as population structure, assortative mating, and intergenerational effects) **[independence assumption, IV2]**
- The maternal genetic variant is not associated with APPOs other than through its association with the maternal risk factor **[exclusion restriction criteria, IV3]**

**Figure 1** Summary of the core Mendelian randomisation assumptions to estimate the presence of an effect of maternal risk factors on adverse pregnancy and perinatal outcomes. IV, instrumental variable.

whether the discrepancy is explained by the different biases. This process can then guide the choice of additional sensitivity analyses or complementary approaches needed to strengthen causal inference.

The Mendelian randomisation in pregnancy (MR-PREG) collaboration was established to improve causal understanding of the effects of maternal lifestyle and health factors on APPOs, as well as efficacy and safety of medication use in pregnancy. In this paper, we outline the aims of the MR-PREG collaboration, describe the characteristics of the contributing studies, detail our approaches to data generation and causal analysis, and summarise findings to date while highlighting ongoing and future work.

### Aims of the MR-PREG collaboration

#### Aim 1: Using triangulation of evidence to improve knowledge on the impact of maternal lifestyle factors on APPOs

The MR-PREG collaboration uses triangulation of genetic methods (eg, MR, and colocalisation) and non-genetic methods (eg, conventional multivariable regression, negative paternal controls, and within-family analyses) to explore the effect of a range of maternal modifiable lifestyle factors, such as smoking, alcohol intake, coffee consumption, adiposity, sleeping habits and physical activity, on the risk of multiple APPOs (figure 2) (online supplemental table 1).[29]

#### Aim 2: Better understanding the effect of maternal predisposition to conditions, such as autoimmune, cardiometabolic, hormonal, reproductive and mental health conditions, on APPOs

An increasing number of women begin pregnancy with one or more pre-existing conditions that may elevate the risk of multiple APPOs.[12] A systematic understanding of the overall impact of these conditions on APPO risk is important for guiding clinical decision-making in the management and treatment of such conditions during pregnancy. Within the MR-PREG collaboration, we employ MR to investigate how genetic predisposition to common conditions in women of reproductive age—such

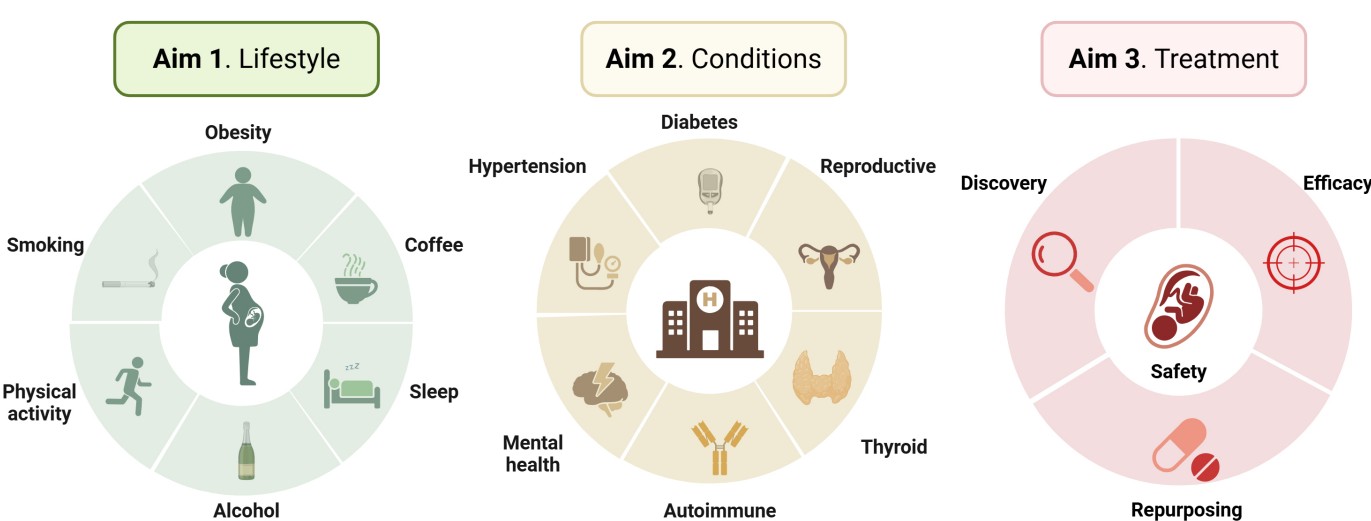

**Figure 2** MR-PREG collaboration aims.
**MR-PREG, Mendelian randomisation in pregnancy.**

as autoimmune, cardiometabolic, hormonal, reproductive and mental health conditions—affects APPO risk (figure 2).

### Aim 3: Leveraging genomics to explore drug efficacy and safety of medications in pregnancy

The MR-PREG collaboration aims to improve the evidence on the benefits and risks of medications during pregnancy using drug target MR. We investigate potential effects of medications for mothers and their babies by (a) investigating the efficacy and safety in pregnancy of medications used to treat pre-existing conditions, (b) discovering new candidate drug targets to prevent/treat APPOs and (c) identifying opportunities to repurpose existing drugs, designed to treat conditions unrelated to pregnancy, to prevent/treat APPOs (figure 2).

## COLLABORATION DESCRIPTION
### Adverse pregnancy and perinatal outcomes (APPOs)

A key focus of the MR-PREG collaboration is on assessing the causal effect of exposures on multiple outcomes simultaneously, so that we have a more complete picture of the potential beneficial, detrimental or null effect of an exposure on APPOs. This 'outcome-wide epidemiology' approach[30] is informative so that women, their partners and healthcare providers have a more balanced view of the potential adverse and beneficial effects of risk factors than what might be obtained from studies that focus on a single or small number of outcomes.

We have included 35 related binary or categorical outcomes that can occur in pregnancy, delivery or the first year postpartum, as well as two underlying continuous traits. Table 1 presents the number of women with both phenotypic and genetic data for each APPO, as well as the number of cases for binary outcomes. APPOs definitions and exclusion criteria are provided in online supplemental table 2A (binary outcomes) and online supplemental table 2B (categorical or continuous outcomes). Study-specific sample sizes for binary and categorical APPOs are reported in online supplemental table 3A (all women with APPO data across core studies), online supplemental table 3B (women with APPO and maternal genetic data across all studies) and online supplemental table 3C (women with APPO and linked fetal genetic data). Study-specific sample size, means and standard deviation (SD) for the continuous traits are reported in online supplemental table 3D and online supplemental table 3E (sample sizes for women with APPO and linked fetal genetic data). Any study-specific deviations from the collaboration definition for APPOs are described in online supplemental tables 4A and B.

### Data sources

Currently, the MR-PREG collaboration includes data from four core studies, which comprise three prospective birth cohorts (the Avon Longitudinal Study of Parents and Children (ALSPAC), Born in Bradford (BiB) and the Norwegian Mother, Father and Child Cohort Study (MoBa)), and a biobank (UK Biobank (UKB)). Data from additional sources are also used, such as genetic association data from publicly available biobanks (FinnGen) and genome-wide association studies (GWAS) meta-analyses, described below.

### Core studies

A brief description of each of the four core studies is presented below and the participant characteristics are summarised in table 2. Online supplemental figures 1-4 show the flow of participants from recruitment into their cohorts to inclusion in our analyses involving maternal genetic and APPOs data. The APPOs each study contributed to MR-PREG are summarised in online supplemental tables 2A and B. Details on genotyping in each study are presented in supplementary material.

### ALSPAC: The Avon Longitudinal Study of Parents and Children

The Avon Longitudinal Study of Parents and Children (ALSPAC) is a prospective birth cohort that started recruiting pregnant women resident in the former county of Avon (centred around the city of Bristol), with delivery dates between April 1991 and December 1992. A total of 14 541 women (ALSPAC-G0) were enrolled during pregnancy (14 676 fetuses) and gave birth to 14 062 live children (ALSPAC-G1).[31 32] Women responded to four questionnaires during pregnancy (average 8, 12, 18 and 32 weeks' gestation) and two postpartum (average 8 weeks and 8 months). Biological samples were collected during pregnancy (blood and urine) and birth (cord blood and placenta). Maternal anthropometrics were based on self-report collected at 12 weeks' gestation and children had anthropometrics measured at birth. Obstetric records were also linked to the participants. Genetic data are available for mothers, partners and children (details on genotyping array and imputation are presented in supplementary material). Children of the children (ALSPAC-G2) are also being assessed and followed-up, but these data have not yet been included in the MR-PREG collaboration. The study website contains details of all the data available through a fully searchable data dictionary and variable search tool: http://www.bristol.ac.uk/alspac/researchers/our-data/. Ethical approval for the study was obtained from the ALSPAC Ethics and Law Committee and the Local Research Ethics Committees. Full details of the ALSPAC consent procedures are available on the study website (http://www.bristol.ac.uk/alspac/researchers/research-ethics/).

### BiB: Born in Bradford

Born in Bradford (BiB) is a prospective birth cohort that recruited women with expected delivery dates between March 2007 and December 2010. Most women were recruited at their oral glucose tolerance test (OGTT) at approximately 26–28 weeks' gestation, which was offered to all women booked for delivery at Bradford Royal Infirmary, except those with known diabetes, during the

**Table 1** Adverse pregnancy and perinatal outcomes (APPOs) and underlying traits, total available sample and respective number of cases

| Type | Outcomes/traits | Total | Cases n (%) |
|---|---|---|---|
| Pregnancy | Hypertensive disorders of pregnancy | 541 768 | 32 549 (6.0%) |
| | Gestational hypertension | 527 932 | 20 777 (3.9%) |
| | Pre-eclampsia | 671 992 | 19 408 (2.9%) |
| | Gestational diabetes | 707 797 | 24 641 (3.5%) |
| | Perinatal depression | 105 656 | 14 591 (13.8%) |
| | Anaemia | 86 167 | 4105 (4.8%) |
| | Pregnancy loss | 298 868 | 94 152 (31.5%) |
| | Miscarriage | 486 217 | 89 086 (18.3%) |
| | Sporadic miscarriage | 241 159 | 55 888 (23.2%) |
| | Recurrent miscarriage | 321 756 | 6233 (1.9%) |
| | Stillbirth | 207 670 | 6331 (3.0%) |
| | Hyperemesis | 478 466 | 4235 (0.9%) |
| | Severity of nausea and vomiting | 82 978 | |
| | Nausea only | | 7166 (8.6%) |
| | Nausea and vomiting | | 30 133 (36.3%) |
| | On medication | | 31 210 (37.6%) |
| Delivery | Induction of labour | 95 239 | 14 495 (15.2%) |
| | Premature rupture of membranes | 310 621 | 21 839 (7.0%) |
| | Caesarean section | 233 969 | 33 909 (14.5%) |
| | Emergency C-section | 90 438 | 9165 (10.1%) |
| | Elective C-section | 87 283 | 6004 (6.9%) |
| | Very pre-term birth | 77 683 | 1107 (1.4%) |
| | Pre-term birth | 285 722 | 18 225 (6.4%) |
| | Spontaneous pre-term birth | 272 940 | 17 176 (6.3%) |
| | Post-term birth | 414 667 | 27 213 (6.6%) |
| | Low birth weight | 283 151 | 19 180 (6.8%) |
| | High birth weight | 266 835 | 6679 (2.5%) |
| | Small for gestational age | 96 305 | 7448 (7.7%) |
| | Large for gestational age | 96 305 | 10 468 (10.9%) |
| | Low Apgar score at 1 min | 86 668 | 4950 (5.7%) |
| | Low Apgar score at 5 min | 74 368 | 821 (1.1%) |
| Postnatal | NICU admission | 77 285 | 6996 (9.1%) |
| | Congenital anomalies | 74 701 | 3569 (4.8%) |
| | Congenital heart disease | 74 701 | 612 (0.8%) |
| | Breastfeeding initiation | 84 668 | 66 408 (78.4%) |
| | Breastfeeding established | 81 258 | 58 035 (71.4%) |
| | Breastfeeding sustained | 65 189 | 44 401 (68.1%) |
| | Breastfeeding duration | 82 429 | |
| | 1 to 3 | | 6372 (7.7%) |
| | 4 to 6 | | 49 123 (59.6%) |
| | >6 | | 3419 (4.1%) |
| Underlying traits | Birth weight* | 289 846 | – |
| | Gestational age* | 189 471 | – |

**Table 1** Continued

| Type | Outcomes/traits | Total | Cases n (%) |
|---|---|---|---|

Details on trait definitions can be found in online supplemental tables 2A and B.
Case and total sample sizes reflect individuals across all data sources contributing to the maternal GWAS meta-analyses. For data sources where sample size varied across genetic variants, we used the median sample size across SNPs as the study-specific sample size.
*Continuous traits.

recruitment period. In BiB, most of the obstetric population consists of women of White British or Pakistani origin (together accounting for 81%, with the remaining women being of other ancestries). A total of 12 453 women (13 776 pregnancies) were enrolled during pregnancy who gave birth to 13 858 live children. Women had anthropometrics measured at recruitment and responded to questionnaires during pregnancy (~28 weeks) and postpartum (6, 12 and 18 months). BiB participants have been linked to maternity health records, hospital admission data and primary care data for both mother and child. Biological samples were collected during pregnancy (blood and urine) and birth (cord blood). Full details of the study methodology were reported previously.[33] Genetic data

are available for mothers and children (details on genotyping array and imputation presented in supplementary material). Ethical approval for the study was granted by the Bradford National Health Service Research Ethics Committee (ref 06/Q1202/48). The study website provides further cohort details and an overview of available data (https://borninbradford.nhs.uk/).

### MoBa: The Norwegian mother, father and child cohort study

The Norwegian mother, father and child cohort study (MoBa) is a prospective birth cohort that recruited pregnant women from all over Norway from 1999 to 2008.[34 35] The cohort includes approximately 95 200 mothers, 75 200 fathers and 114 500 children. Mothers and their partners

**Table 2** Maternal participant characteristics in the participating core studies*

| Characteristics | ALSPAC | BiB | MoBa | UK Biobank |
|---|---|---|---|---|
| Total N | 13 895 | 9407 | 93 512 | 224 423 |
| Age, mean (SD) | 28.0 (5.0) | 27.4 (5.6) | 30.1 (4.7) | 25.3 (4.6)† |
| Missing | 369 | 1169 | 458 | 45 402 |
| Ethnicity (White), n (%) | 11 600 (97.4%) | 3430 (40.9%) | 88 989 (97.3%) | 211 677 (94.6%) |
| Missing | 1982 | 1025 | 2021 | 706 |
| Education (university level), n (%) | 1544 (12.9%) | 1432 (17.4%) | 50 047 (60.7%) | 65 334 (29.5%)‡ |
| Missing | 1890 | 1186 | 11 064 | 2714 |
| Index of multiple deprivation, n (%) | | | | |
| 1st quintile (most deprived) | 1628 (13.9%) | 5232 (63.5%) | NA | NA |
| 5th quintile (least deprived) | 3045 (26.0%) | 158 (1.9%) | NA | NA |
| Missing | 2197 | 1171 | | |
| Parity (nulliparous), n (%) | 5619 (44.7%) | 3768 (41.6%) | 42 135 (45.3%) | NA§ |
| Missing | 1333 | 345 | 399 | |
| Smoking during pregnancy (ever), n (%) | 3301 (25.9%) | 1408 (17.1%) | 5939 (7.7%) | NA |
| Missing | 1124 | 1185 | 16 824 | |
| Alcohol intake during pregnancy, n (%) | 8036 (64.0%) | 1707 (23.4%) | 2472 (3.8%) | NA |
| Missing | 1340 | 2114 | 27 648 | |
| Body Mass Index¶, mean (SD) | 22.9 (3.8) | 26.1 (5.7) | 24.1 (4.3) | NA‡ |
| Missing | 2711 | 1562 | 11 561 | |

This includes all available data at the outcome level. Percentages for each category are calculated using the total number of participants with non-missing data as the denominator.
*Participants included if they had data on at least one APPO.
†Age at first birth.
‡ Information assessed at recruitment and not at pregnancy.
§ In UKB, the sample was restricted to women who have ever been pregnant.
¶ Assessed pre-pregnancy or during pregnancy.
ALSPAC, The Avon Longitudinal Study of Parents and Children; BiB, Born in Bradford; MoBa, The Norwegian Mother, Father and Child Cohort Study; NA, information not available in this study.

responded to questionnaires during pregnancy (15, 22 and 30 weeks' gestation) and at multiple times post-partum (starting from when the child was 6 months old). Blood samples were obtained from both parents during pregnancy and from mothers and children (cord blood) at birth. Maternal anthropometrics were collected from a questionnaire at 15 weeks' gestation. Data were linked to the Medical Birth Registry (MBRN), which is a national health registry containing information about all births in Norway. Genetic data are available for mothers, children and partners (details on genotyping array and imputation presented in supplementary material). The establishment of MoBa and initial data collection was based on a licence from the Norwegian Data Protection Agency and approval from The Regional Committees for Medical and Health Research Ethics. The MoBa cohort is currently regulated by the Norwegian Health Registry Act. Ethical approval for our study was obtained from The Regional Committees for Medical and Health Research Ethics (ref 2018/1256).

### UKB: UK Biobank

UK Biobank (UKB) is an adult cohort that retrospectively collected relevant data on APPOs. All people in the UK National Health Service (NHS) registry aged between 40 and 69 years and living within an approximately 25-mile radius from one of the 22 study centres were invited to participate in UKB between 2006 and 2010.[36 37] A total of 500 000 adults (5.5% of the ~9.2 million invited) were recruited into the study (54.4% females). Information was assessed at baseline via a self-completed questionnaire, physical measures (including anthropometrics), and collection of non-fasting blood, urine and saliva. Participants have been followed up by linkage to electronic health records and a subset of participants responded to online questionnaires. Hospital Episode Statistics (HES) from 1997 for England, 1998 for Wales and 1981 for Scotland are available.[38] HES data also contain maternity-related admissions for England and Wales. It is important to note that not every participant has a hospital inpatient record, as not all have been admitted to hospital within the period covered. Genetic data from UKB participants are available (details on genotyping array and imputation presented in supplementary material). Ethical approval for UKB was obtained from the Northwest Multi-Centre Research Ethics Committee (MREC), and MR-PREG collaboration studies are linked to UKB application number 23938. The UKB showcase website contains details of the data available: https://biobank.ndph.ox.ac.uk/showcase/.

### Additional data sources

To increase statistical power for MR analyses, the MR-PREG collaboration also uses genetic association data for APPOs from publicly available datasets—ie, FinnGen and several publicly available GWAS meta-analyses—as described below.

### FinnGen

FinnGen is a nationwide network of Finnish biobanks linked to national electronic health registries that provide information on prescriptions and disease diagnoses (International Classification of Diseases (ICD)-9 and ICD-10 codes).[39] The current FinnGen release (R12) includes data from 500 348 individuals (282 064 females). In addition to clinical endpoints, including several APPOs, genetic data are also available. The APPOs and corresponding numbers of cases and controls contributed by FinnGen are provided in online supplemental table 3B. FinnGen has been approved by the Coordinating Ethics Committee of the Helsinki and Uusimaa Hospital District (Nr HUS/990/2017). Further information on FinnGen is available at https://www.finngen.fi/en. The metadata used by the MR-PREG collaboration can be accessed at https://www.finngen.fi/en/access_results.

### Publicly available GWAS meta-analyses

At the time of writing, the MR-PREG collaboration has harmonised and quality-controlled data from GWAS meta-analyses for GDM (5485 cases and 347 856 controls from the GENetics of Diabetes in Pregnancy Consortium (GenDIP) consortium[40]; PE (9515 cases and 157 719 controls from the International Pregnancy Genetics (InterPregGen) consortium[41]; gestational duration-related traits (18 797 PTB cases and 260 246 controls, 15 972 post-term birth cases and 115 307 controls, and 195 555 individuals with gestational age at delivery from the Early Growth Genetics (EGG) consortium[42]; post-natal depression (17 339 cases and 53 426 controls from the Psychiatric Genomics consortium (PGC)).[43] Details are provided in online supplemental table 5, including the number of participants, outcome definition and source of data (eg, electronic health records, maternal report, research data collection), and availability of maternal and/or fetal genetic effects.

### Genetic association data for adverse pregnancy and perinatal outcomes

For studies with access to individual-level data (ALSPAC, BiB, MoBa and UKB), we conducted GWAS analyses to generate genetic association data for APPOs, enabling two-sample MR analyses across the aims of the MR-PREG collaboration. The procedures used for conducting GWAS and quality control in each study are described in detail in the online supplemental text. We excluded genetic variants with low imputation accuracy (INFO score <0.4) and/or low minor allele frequency (MAF <0.01). In addition, we excluded studies that overlapped with public GWAS meta-analyses or that contributed fewer than 50 cases for a given APPO and applied genomic control to each study. We then pooled study-specific genetic association data on APPOs using inverse variance weighted, fixed-effects meta-analyses implemented in METAL (v. 2020-05-05)[44] and estimated Cochrane's Q statistic to explore between-study

| Relevance | Independence | Exclusion Restriction |
|---|---|---|
| **Assumption: The maternal genetic instrument is robustly associated with the maternal exposure during pregnancy** <br><br> • Many instruments for an exposure are based on non-pregnant phenotypes (e.g., BMI, smoking, alcohol use, or sleep measured outside of pregnancy) rather than pregnancy-specific measures or measures taken during pregnancy <br><br> • Genetic variants are lifelong so may not easily capture exposures in specific time frames like pregnancy which bring about widespread physiological and hormonal changes <br><br> **Recommendations** <br><br> • We recommend ensuring a strong association between the maternal genetic variant and the pregnancy-specific exposure | **Assumption: There is no confounding of the maternal genetic variant(s) and the outcome (i.e., population level confounders)** <br><br> • Conventional confounding by socioeconomic position and its related lifestyle and health characteristics cannot confound MR analyses as these cannot cause variation in genetic variants <br><br> • Population stratification, assortative mating, intergenerational effects can introduce confounding in genetic studies <br><br> **Recommendations** <br><br> • We account for population stratification in each study by adjusting for ancestral principal components or using mixed models <br><br> • Family based designs can avoid bias by population stratification, assortative mating, intergenerational effects | **Assumption: The maternal genetic variant is not associated with the outcome other than through its association with the maternal exposure** <br><br> • One plausible violation of this assumption could be horizontal pleiotropy i.e. where the genetic instrument influences the outcome not through the exposure of interest <br><br> • Estimates can also be biased if genetic variants which influence the maternal exposure also influence the outcome when inherited by the offspring (**Figure 3a**) <br><br> **Recommendations** <br><br> • Ideally researchers should consider accounting for fetal genotype in MR of maternal exposures <br><br> • Adjusting only for offspring genotype may cause collider bias or reflect assortative mating (**Figure 3b**) <br><br> • We recommend further adjusting for paternal genotype in sensitivity analyses for later offspring outcomes, though sample sizes may be smaller |

**Figure 3** Approaches to explore the plausibility of the Mendelian randomisation core assumptions (figure 1), with a focus on the context of maternal risk factors and adverse pregnancy and perinatal outcomes (APPOs).
For a review of standard MR sensitivity analyses, we refer readers to Sanderson et al.[61]
BMI, body mass index; MR, Mendelian randomisation.

heterogeneity. Separate meta-analyses were conducted for maternal and offspring genetic effects. We used GWASInspector to evaluate the quality of both study-level and meta-analytic GWAS summary data.[45]

## Decomposing maternal and fetal genetic contributions to APPOs

The MR-PREG collaboration is primarily interested in causal effects of maternal exposures on APPOs. Due to the correlation between maternal and offspring genotype, accounting for offspring genetic effects is crucial for reliable interpretation of MR findings of maternal exposure effects (figure 3, figure 4a and figure 4b). Genetically instrumented maternal exposure effects (not biased due to exclusion restriction violation via fetal effects) and genetically instrumented fetal effects (not confounded by maternal genotype) can be estimated. We used a weighted linear model (WLM) implemented in the DONUTS (Decomposing nature and nurture using GWAS summary statistics) software[46] to estimate mutually adjusted maternal and fetal genetic

effects at each single nucleotide polymorphism (SNP) for each APPO. The WLM calculates conditional genetic effects on APPOs (ie, the mutually adjusted coefficients for maternal, offspring and paternal genotype, fitted jointly in the same model) as linear combinations of the marginal genetic effects (ie, the coefficients for maternal, offspring and paternal genotype, estimated separately in potentially overlapping samples). We created two sets of conditional genetic effects estimates on APPOs. The first corresponded to mutually adjusted maternal and fetal estimates, leveraging data from the maternal and fetal GWAS meta-analyses. The second corresponded to mutually adjusted maternal, fetal and paternal estimates, leveraging data from maternal and fetal GWAS meta-analyses, as well as paternal GWAS conducted in MoBa. Details can be found in online supplemental text.

## PATIENT, PARTICIPANT AND PUBLIC INVOLVEMENT (PPI)

The birth cohorts involved in this collaboration (ALSPAC,[47] BiB[33] and MoBa)[35] have a strong track record

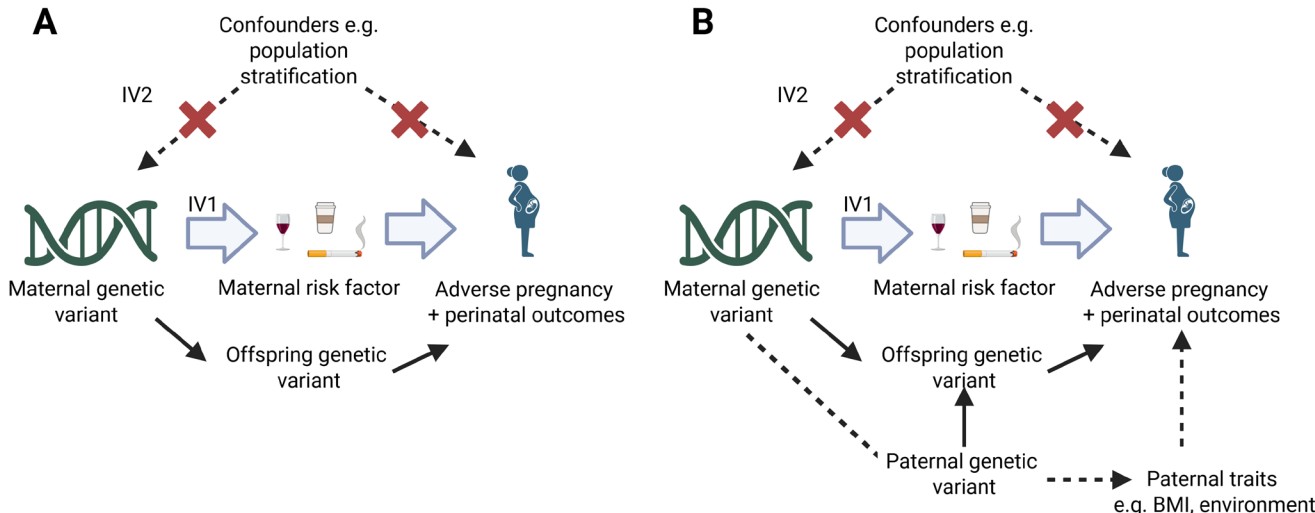

**Figure 4** (A) Illustration of how the exclusion restriction assumption may be violated. (B) Illustration of how collider bias might be induced in MR analysis when conditioning on offspring genetic variants.

**Offspring genetic variation might violate the exclusion restriction assumption (A). Adjusting for offspring genetic variants only (which are collider variables) (B) means that a false association between maternal and paternal genetic variants can be created (shown as a dashed line). Such associations can also arise via assortative mating. This will only cause bias if the outcome is influenced by paternal genetic variants, independently of maternal and offspring genetic variants. However, by additionally adjusting for paternal genetic variants, the pathway highlighted in black dashed arrows can be closed.**

**BMI, body mass index; IV, Instrumental variable; MR, Mendelian randomisation.**

of patient, participant and public involvement (PPI). For example, in ALSPAC, since 2021, participants who have become pregnant, or a parent, and their children have been invited to be involved in the ALSPAC-G2 cohort.[48] PPI with them identified new data collection that has informed this collaboration, such as understanding the potential effects of vaping, mental health and climate change on APPOs. Similarly, the current offspring of BiB, although young for starting pregnancy (currently aged 14 to 18), have been involved in designing the current follow-up that will go into their early 20s, and highlighted reproductive health as an important area for research. The design of the MoBa study, the recruitment procedures, the aims of the study and the research questions were discussed during the 1990s with the Norwegian Society for Gynecology and Obstetrics, international colleagues and at national meetings with midwives. The study was also discussed in the media and in the Norwegian Parliament before it became a national study.

### Findings to date

To date, there have been 12 peer-reviewed publications[23] [49–59] and one preprint[60] resulting from the MR-PREG collaboration. We briefly describe a selection of these here to give readers an indication of the different ways in which MR-PREG can be used to explore effects of exposures on APPOs.

Most publications to date align with Aim 1, enhancing our understanding of how maternal lifestyle factors influence the risk of APPOs, thereby informing interventions aimed at their prevention. For example, by triangulating evidence from MR, conventional multivariable regression and paternal negative control, we have shown that higher maternal body mass index (BMI) increases the risk of GH, PE, GDM, pre-labour membrane rupture, induction of labour, caesarean section, LGA, high birth weight, low Apgar score and admission to a neonatal intensive care unit. We also found evidence of higher maternal BMI reducing the odds of SGA and breastfeeding and having no detectable effect on perinatal depression.[52] Triangulating evidence from MR and conventional multivariable regression, we found evidence that insomnia may increase the risk of perinatal depression but does not appear to influence most other APPOs that we were able to explore.[54] In a study focused on fetal growth trajectories, assessed by repeat ultrasound scan measures in two cohorts, we triangulated evidence from MR, conventional multivariable regression and paternal negative control study and found evidence of a consistent linear dose-response association of maternal smoking with fetal growth from early in the second trimester onwards. No major growth deficit was found in women who quit smoking early in pregnancy.[57]

In relation to Aim 3, we have published preliminary work identifying novel molecular targets (ie, proteins and metabolites) causally related to APPOs. In one study, we used MR to investigate the effect of 1139 maternal and fetal circulating proteins on offspring birth weight. Results support maternal and fetal protein effects on birth weight, implicating roles for glucose metabolism,

energy balance and vascular function.[60] For example, we found evidence that higher maternal levels of Proprotein Convertase Subtilisin/Kexin Type 1 (PCSK1) potentially increase birth weight while higher maternal levels of Galectin-4 (LGALS4) potentially decrease birth weight. In contrast, fetal genetically predicted PCSK1 and LGALS4 showed effects in the opposite direction. We have also identified maternal circulating metabolites beyond glucose that may influence birth weight, such as amino acids, including glutamine.[49] In further studies, we used conventional multivariable regression and MR to explore the potential effect of >1000 maternal circulating metabolites on the risk of offspring congenital heart disease. We found that pregnancy amino acid metabolism, androgenic steroid lipids and levels of succinylcarnitine may be contributing factors for congenital heart disease.[53]

### Strengths and Limitations

The collaboration has created curated data for 35 APPOs and harmonised data across several studies to support large-scale investigations of causes of APPOs. In addition, the generation of genetic association data enables well-powered genetic studies, including MR studies, to improve causal knowledge of many key targets for lifestyle and pharmaceutical interventions aimed at preventing APPOs. Data on equivalent exposures in partners and on a wide range of plausible confounders enable triangulation across genetic and non-genetic approaches, and also pertinent sensitivity analyses. In relation to those sensitivity analyses, the collaboration has also derived conditional estimates so that maternal genetic variants can be used as an instrument to test the effect of maternal exposures without potential biases related to offspring genetic effects.

Despite the size and scale of the data, we are still underpowered to detect causal effects on rare APPOs, such as congenital anomalies and low Apgar scores. Furthermore, participants are of predominantly European ancestry, except for BiB. Enhancing the ancestry diversity in the data contributing to the collaboration is a key priority as outlined below in 'Collaborations and future plans'. Finally, despite our best efforts to derive accurate and standardised APPO definitions, there is inevitably a considerable degree of misclassification, especially where some studies derived their data only from self-reported information or only from electronic health records.

### Collaborations and future plans

We are currently focussing on systematically assessing the effects of predisposition to autoimmune, mental health, thyroid, reproductive and cardiometabolic disorders on APPOs (Aim 2), discovering new candidate drug targets for APPOs (Aim 3), and investigating the efficacy, safety and potential of repurposing existing drugs during pregnancy (Aim 3).

A future priority is to increase diversity in the genetic background of MR-PREG collaboration participants, given the current effort is predominantly focussed on participants of European ancestry. This will enable us to improve the internal validity of the MR-PREG studies (eg, leveraging trans-ancestral data to mitigate challenges due to linkage disequilibrium) but also external validity (eg, testing transportability of estimated effects across ancestries).

Additionally, we are incorporating more molecular data into MR-PREG studies to better understand the molecular mechanisms underlying APPOs, for example, placental transcriptomics and proteomics. To enhance our work identifying molecular targets for the prevention and treatment of APPOs (Aim 3), we are using whole-exome and whole-genome sequencing data to pinpoint causal genes linked to these adverse outcomes.

We are advancing collaborations with additional large-scale studies including genetic and APPO data, with a focus on rarer outcomes such as congenital anomalies and participants of non-European ancestry to increase statistical power for key safety outcomes and ancestry diversity.

**Author affiliations**
[1]MRC Integrative Epidemiology Unit at the University of Bristol, Bristol, UK
[2]Population Health Sciences, Bristol Medical School, University of Bristol, Bristol, UK
[3]Department of Endocrine and Metabolic Diseases, Shanghai Institute of Endocrine and Metabolic Diseases, Ruijin Hospital, Shanghai Jiao Tong University School of Medicine, Shanghai, China
[4]Shanghai National Clinical Research Center for Metabolic Diseases, Key Laboratory for Endocrine and Metabolic Diseases of the National Health Commission of the PR China, Shanghai Key Laboratory for Endocrine Tumor, Lifecycle Health Management Center, Ruijin Hospital, Shanghai Jiao Tong University School of Medicine, Shanghai, China
[5]Frazer Institute, University of Queensland, Woolloongabba, Queensland, Australia
[6]Division of Surgery and Interventional Science, University College London, London, UK
[7]Population Health, School of Medicine and Population Health, University of Sheffield, Sheffield, UK
[8]Bradford Institute for Health Research, Bradford Teaching Hospitals, Bradford, UK
[9]Centre for Fertility and Health, Norwegian Institute of Public Health, Oslo, Norway

**Acknowledgements** The authors would like to thank Dr Marwa Al-Arab, Dr Alba Fernandez-Sanles, Dr Helena Urquijo, Jevvy Huang and Dr Peiyuan Huang for their assistance with data preparation and statistical analysis. The authors also thank Professor Dave Evans for the valuable discussions regarding the implementation of WLM models to estimate conditional genetic effects. All cohort-specific acknowledgements are detailed in the online supplemental file 1.

**Contributors** The MR-PREG collaboration was conceptualised by DAL and MCB. GC, NM, AGS, MCB, QY, TB, CC, EA and AT undertook data curation and generated GWAS pipelines for the individual cohorts. QY and MCB ran the meta-analysis. TB and MCB generated the WLM models. GC, NM, MCB, AGS and DAL wrote the paper, and all authors edited and approved the manuscript. This publication is the work of the authors and MCB acts as guarantors for the contents of this paper.

**Funding** All cohort-specific funding information is detailed in the online supplemental file 1. DAL, MCB, GC, AGS, TB, QY, AT, HC, EA, PH and NM work in a unit supported by the University of Bristol and UK Medical Research Council (MC_UU_00032/5) and the British Heart Foundation (AA/18/1/34219). AGS was supported by the European Union's Horizon 2020 research and innovation programme (grant agreement No 874739, LongITools). AGS, GC and DAL are supported by STAGE that has received funding from the European Union's; Horizon Europe Research and Innovation Programme under grant agreement nº 101137146 (via UKRI grant number 10112787 (Beta Technology) 10099041). EA is supported by a Wellcome Trust PhD studentship (228276/Z/23/Z). MCM is supported by the Research Council of Norway through its Centres of Excellence funding scheme (project No. 262700) and the European Research Council under the European Union's Horizon 2020 research and innovation programme (ERC Starting Grant,

INFERTILITY grant agreement No. 947684). JW was funded for BiB data collection by a UK Medical Research Council (MRC) Special Training Fellowship in Health of the Public and Health Services Research (MRC GO601712) and is now supported by supported by the UK Prevention Research Partnership (MR/S037527/1), an initiative funded by UK Research and Innovation Councils, the Department of Health and Social Care (England) and the UK devolved administrations, and leading health research charities.

**Competing interests**  None declared.

**Patient and public involvement**  Patients and/or the public were involved in the design, or conduct, or reporting, or dissemination plans of this research. Refer to the Methods section for further details.

**Patient consent for publication**  Not applicable.

**Ethics approval**  Ethical approval for the study was obtained from the ALSPAC Ethics and Law Committee and the Local Research Ethics Committees. Full details of the ALSPAC consent procedures are available on the study website (http://www.bristol.ac.uk/alspac/researchers/research-ethics/http://www.bristol.ac.uk/alspac/researchers/research-ethics/). Ethical approval for the study was granted by the Bradford National Health Service Research Ethics Committee (ref 06/Q1202/48). The study website provides further cohort details and an overview of available data (https://borninbradford.nhs.uk/). The establishment of MoBa and initial data collection was based on a licence from the Norwegian Data Protection Agency and approval from The Regional Committees for Medical and Health Research Ethics. The MoBa cohort is currently regulated by the Norwegian Health Registry Act. Ethical approval for our study was obtained from The Regional Committees for Medical and Health Research Ethics (ref 2018/1256). Ethical approval for UKB was obtained from the Northwest Multi-Centre Research Ethics Committee (MREC), and MR-PREG collaboration studies are linked to UKB application number 23938. The Coordinating Ethics Committee of the Helsinki and Uusimaa Hospital District has approved the FinnGen consortium (Nr HUS/990/2017). More information about FinnGen can be found on the website https://www.finngen.fi/en. The metadata from FinnGen used by the MR-PREG collaboration is publicly available at https://www.finngen.fi/en/access_results. Participants gave informed consent to participate in the study before taking part.

**Provenance and peer review**  Not commissioned; externally peer reviewed.

**Data availability statement**  Data may be obtained from a third party and are not publicly available. Analytical code used in the MR-PREG collaboration can be accessed at https://github.com/MRCIEU/MR-PREG. Genetic association data for APPOs generated by the MR-PREG collaboration can only be used for research that is covered by data agreements with current contributing studies. The ALSPAC access policy that describes the proposal process in detail including any costs associated with conducting research at ALSPAC, which may be updated from time to time, and is available at: https://www.bristol.ac.uk/medialibrary/sites/alspac/documents/researchers/dataaccess/ALSPAC_Access_Policy.pd. Data are available upon request from BiB, and information is available at: https://borninbradford.nhs.uk/research/how-to-access-data/. Data from MoBa are available upon application to Helsedata administered by the https://borninbradford.nhs.uk/research/how-to-access-data/. Norwegian Institute of Public Health (see its website text=Relevant%20information%20for%20researchers%20applying%20for%20acc ess%20to,be%20submitted%20via%20the%20application%20form%20on%20helsedata.no.for details). Researchers can apply for access to the UK Biobank data via the Access Management System (AMS) (https://www.ukbiobank.ac.uk/enable-your-research/apply-for-access).

**ORCID iDs**
Gemma L Clayton https://orcid.org/0000-0002-9525-2758
Ana Goncalves Soares https://orcid.org/0000-0003-2763-4647
Tom A Bond https://orcid.org/0000-0002-9298-6860
Elisabeth Aiton https://orcid.org/0000-0002-0001-3480
Jane West https://orcid.org/0000-0002-5770-8363
Maria Carolina Borges https://orcid.org/0000-0001-7785-4547

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
