## [Reviewer comments · BMJ Open]

ARTICLE DETAILS

Title (Provisional)

Cohort Profile: The Mendelian Randomization in Pregnancy (MR-PREG) collaboration - Improving evidence for prevention and treatment of adverse pregnancy and perinatal outcomes

Authors

McBride, Nancy; Clayton, Gemma; Soares, Ana Luiza Goncalves; Yang, Qian; Bond, Tom A; Taylor, Amy; Chatzigeorgiou, Charikleia; Aiton, Elisabeth; West, Jane; Magnus, Maria C; Lawlor, Deborah; Borges, Maria Carolina

VERSION 1 - REVIEW

Reviewer	1
Name	Kjaergaard, Alisa
Affiliation	Aarhus University Hospital, Steno Diabetes Center Aarhus
Date	13-Oct-2025
COI	None

This manuscript presents a cohort profile describing the Mendelian Randomization in Pregnancy (MR-PREG) collaboration.

The Introduction section would benefit from the use of subheadings (e.g., "Aims of the MR-PREG Collaboration"), which would help break the text into clear, logical sections and improve readability.

On page 10, lines 7–12, there is a particularly long and complex sentence that would be clearer if simplified and rephrased.

Reviewer	2
Name	Kattah, Andrea
Affiliation	Mayo Clinic
Date	26-Nov-2025
COI	None

This is a lovely description of a cohort that is designed to assess the associations of exposures (lifestyle, comorbidities and drugs) and APPOs. The authors do a great job of describing Mendelian randomization and why it can be useful in this context. The discussion of preliminary findings is helpful. I agree that recruiting some non-white populations is critical.

VERSION 1 - AUTHOR RESPONSE

Reviewer: 1

Dr. Alisa Kjaergaard, Aarhus University Hospital

Comments to the Author:

This manuscript presents a cohort profile describing the Mendelian Randomization in Pregnancy (MR-PREG) collaboration.

The Introduction section would benefit from the use of subheadings (e.g., “Aims of the MR-PREG Collaboration”), which would help break the text into clear, logical sections and improve readability.

Response: Thank you for this suggestion. We have now included subheadings in the introduction, which will hopefully improve readability. The subheadings included are:

- Why was the collaboration established?
- Aims of the MR-PREG collaboration
- Collaboration description

On page 10, lines 7–12, there is a particularly long and complex sentence that would be clearer if simplified and rephrased.

Response: We have reduced and rephrased slightly the sentence. It now reads as:

“Triangulation of evidence acknowledges and exploits the fact that all methods have sources of bias²⁹. It involves integrating multiple lines of evidence using one or more different approaches (e.g., different analytical methods, different data sources, or different study designs) with distinct and unrelated key sources of bias. If results are consistent, this increases the credibility of that being the correct causal effect, as it would be unlikely for different biases to produce similar results.”

Reviewer: 2

Dr. Andrea Kattah, Mayo Clinic

Comments to the Author:

This is a lovely description of a cohort that is designed to assess the associations of exposures (lifestyle, comorbidities and drugs) and APPOs. The authors do a great job of describing Mendelian randomization and why it can be useful in this context. The discussion of preliminary findings is helpful. I agree that recruiting some non-white populations is critical.

Response: Thank you for the positive feedback on our manuscript.